

# Reproductive biology of little tunny *Euthynnus alletteratus* (Rafinesque, 1810) in the southwest Gulf of Mexico

Roberto Cruz-Castán[1,2], César Meiners-Mandujano[2], David Macías[3], Lourdes Jiménez-Badillo[2] and Sergio Curiel-Ramírez[4]

[1] Posgrado en Ecología y Pesquerías, Universidad Veracruzana, Boca del Río, Veracruz, Mexico
[2] Instituto de Ciencias Marinas y Pesquerías, Universidad Veracruzana, Boca del Río, Veracruz, Mexico
[3] Centro Oceanográfico de Málaga, Instituto Español de Oceanografía, Fuengirola, Malaga, Spain
[4] Instituto de Investigaciones Oceanológicas, Universidad Autónoma de Baja California, Ensenada, Baja California, Mexico

## ABSTRACT

The aim of this study was to describe the reproductive dynamic of *Euthynnus alletteratus* in the southwest Gulf of Mexico. The annual variation of the volume fraction occupied by gametes and tissues in gonads were related with main body indexes, such as the gonadosomatic index ($I_G$), the hepatosomatic index ($I_H$), and the nutrition index ($I_N$), and compared with the sea surface temperature. A total of 951 *E. alletteratus* individuals were sampled, where a sex ratio of 1:1 and a size interval strongly skewed towards organisms with a fork length ($L_F$) of 36–40 cm were observed. The $I_G$ showed an increase from March to September with maximum values in April and July. Two clearly defined peaks were observed, and they were consistent with the histological analysis, where the percentage of ripe gametes predominated from April to September. The optimum thermal window for reproductive activity was from 24 to 28 °C. The size of first sex maturity was 34.35 cm of $L_F$ for males and 34.60 ($L_F$) for females, without significant difference between sexes.

## INTRODUCTION

The little tunny (*Euthynnus alletteratus* (*Rafinesque, 1810*)) is an epipelagic and neritic fish (*Chur, 1973*) distributed on both sides of the tropical and subtropical Atlantic Ocean, including the Mediterranean Sea, the Black Sea, the Caribbean Sea, and the Gulf of Mexico (*Belloc, 1955*; *Collette & Nauen, 1983*; *Valeiras & Abad, 2006*). It is the smallest of the tunids and grows to a maximum weight of 16.5 kg and a total length of 122 cm fork length ($L_F$) (*Froese & Pauly, 2014*). As a target species, it is historically valued and captured seasonally by coastal trawling fleets in western Africa, the Gulf of Cadiz, and the Mediterranean Sea (*Sabatés & Recasens, 2001*; *Neves dos Santos & García, 2006*; *Gaykov & Bokhanov, 2008*; *Kahraman, 2005*; *Zengin & Karakulak, 2009*). It is also captured incidentally in important quantities by industrial trawling and purse seine fleets between Mauritania and Angola (*Gaykov & Bokhanov, 2008*). In the southern Gulf of Mexico,

Corresponding author
Sergio Curiel-Ramírez,
curiels@uabc.edu.mx

*E. alletteratus* is considered a secondary target species and captured mainly by the multispecific artisanal fishery for the local market and as bait (*Cabrera et al., 2005*). For its capture, different fishing gears such as gillnets, drift nets, surface long-lines, and hand lines with bait or trolling are used, depending on the locality (*Cabrera et al., 2005*; *Jiménez-Badillo et al., 2006*).

Currently, direct or incidental capture of little tunny developed in the Gulf of Mexico is not subject to fishing control or specifications, and there are no estimates of the stock size of this species in the Gulf of Mexico. Information about its structure population is scarce and fragmented, which prevents adequate assessment of the stock (*Valeiras & Abad, 2006*). The conservation and management strategies of tuna stocks require updated and specific information about their reproductive biology; this allows the establishment of exploitation criteria that contribute to adequate management scenarios for this species.

Most studies about the biology of *E. alletteratus* have taken part along its eastern distribution, referring to the size at first maturity, body indexes, fecundity, sex ratio, spawning periods, age and growth, somatic growth, size structure, and trophic ecology (*Frade & Postel, 1955*; *Landau, 1965*; *Rodríguez-Roda, 1966*, *1979*; *Diouf, 1981*; *Cayré & Diouf, 1983*; *Kahraman & Oray, 2001*; *Kahraman, 2005*; *Macías et al., 2006*; *Bahou et al., 2007*; *Falutano et al., 2007*; *Kahraman et al., 2008*; *Macías et al., 2009*; *Hajjej et al., 2010a*). In contrast, and despite of its extensive geographical distribution and regional fisheries importance, the biology of *E. alletteratus* has been studied scarcely along the western Atlantic margin (*Manooch, Manson & Nelson, 1985*; *Cabrera et al., 2005*; *García & Posada, 2013*).

Fish reproduction is a complex physiological process, markedly seasonal. It is a result of environmental information incorporated by the fish and transmitted by hormones, with the objective of reproducing only in the most favorable temporal and spatial window to maximize the survival of the progeny (*Muñoz et al., 2005*). In spite of the fact that *E. alletteratus* is a conspicuous component in artisanal fisheries of the southwest Gulf of Mexico throughout the year, there is no scientific and updated information about its reproductive biology in this area. Therefore, the aim of this study was to describe the reproductive biology of *E. alletteratus* fished in the southwest Gulf of Mexico, by analyzing their sex ratio, the size at first maturity and the reproductive cycle with the use of body indexes and the annual variation of the volume fraction occupied by gametes in gonads. Furthermore, we examined the relationship between sea surface temperature (SST) and reproductive activity by months.

## MATERIALS AND METHODS
### Sampling
From December 2009 to November 2012, artisanal commercial catch-based samplings of *E. alletteratus* were conducted in the southwest Gulf of Mexico, particularly for the area between the north of the Veracruz Reef System and Punta Roca Partida, in the Mexican state of Veracruz (Fig. 1). Sampling periodicity varied between four and twelve weeks, according to availability of *E. alletteratus* captures that were landed in Antón Lizardo and Las Barrancas at the municipality of Alvarado. The collected organisms were transported to the laboratory for biological processing.
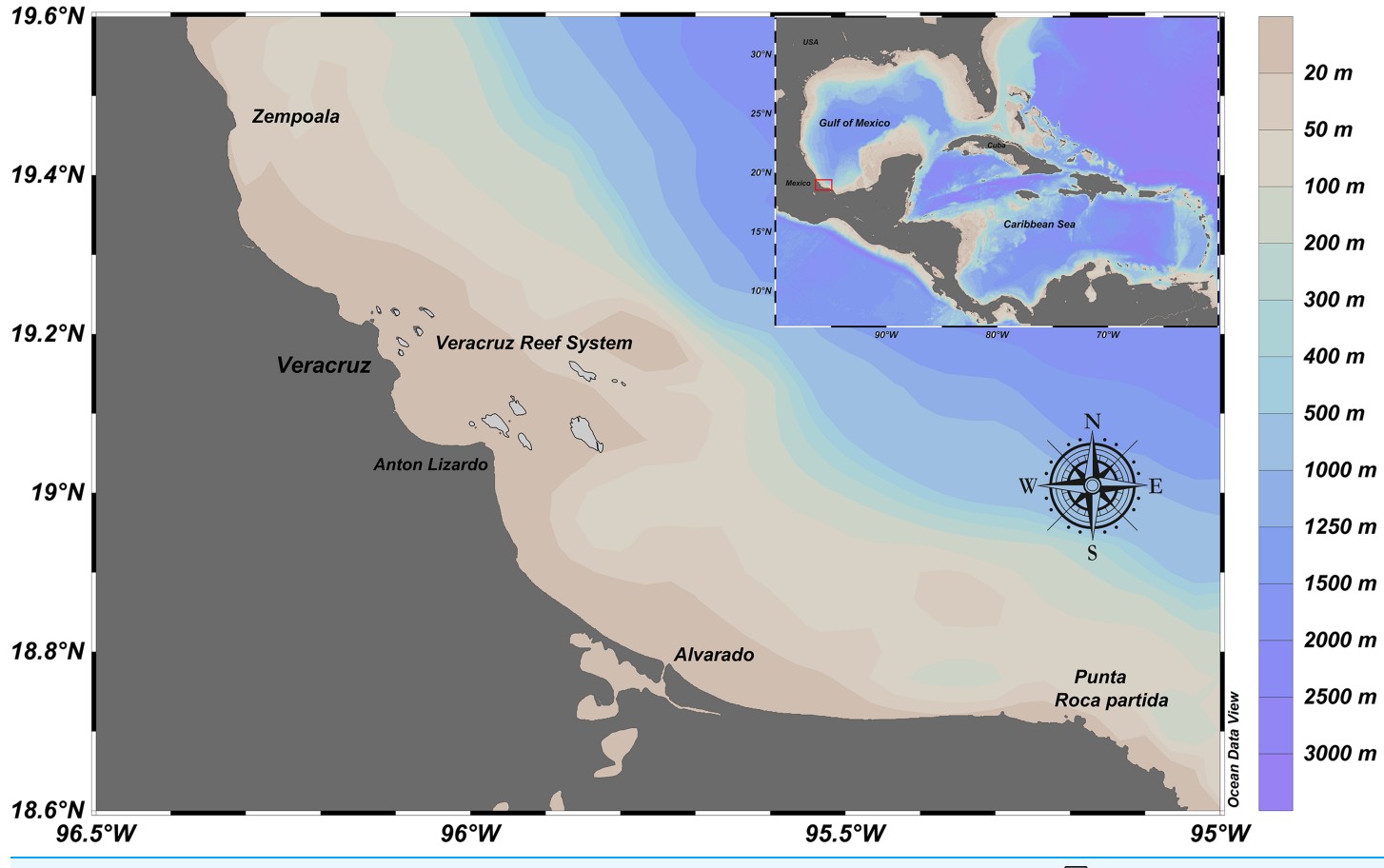

**Figure 1 Fishing ground location of *E. alletteratus* caught in the southwest Gulf of Mexico.**

## Macroscopic analysis

Each specimen was measured at 0.1 cm closer to $L_F$ and weighed (±0.1 g). The sex, and weight of the gonads and livers (±0.001 g) were registered, and the sex maturity was determined macroscopically within six stages according to the characteristics observed in gonads (Table 1).

To obtain size structure, the data were grouped in size classes of intervals of five cm of $L_F$, then the sex ratio was calculated by size class and coefficient of variation (CV) through nonparametric bootstrap using the library "inbio 1.2" (*Sampedro, Trujillo & Saínza, 2005*) in R software (*R Core Team, 2013*); additionally, a chi-squared test was performed to determine if the sex ratio is equal.

Gonadosomatic index ($I_G$) and the hepatosomatic index ($I_H$) were calculated according to the relation of the weight of the gonad and the liver regarding to the total weight of each organism, as described in *Introzzi & De Introzzi (1986)*:

$$I_G = \frac{\text{Gonad weight}}{\text{Total weight} - \text{Gonad weight}} \cdot 100$$

$$I_H = \frac{\text{Liver weight}}{\text{Total weight} - \text{Liver weight}} \cdot 100$$

**Table 1 Macroscopic scale to determine the maturity in the gonads of *E. alletteratus* caught from December 2011 to November 2012 in the southwest Gulf of Mexico, adapted from *Bezerra et al. (2013)*.**

| Maturity status | Activity | Gonadal phase | Female | Male |
|---|---|---|---|---|
| Immature | Inactive | Immature | • Small ovaries<br>• White to pink color<br>• Ovaries only located in a small part at the abdominal cavity | • Tiny testes and translucent<br>• Located in a small part at abdominal cavity |
| Immature | Inactive | Maturation | • Ovaries white to pink color<br>• Ovaries larger than in immature phase with more solid consistency and increasing in volume<br>• Oocytes are not distinguished yet | • Testes located in half abdominal cavity<br>• White testes<br>• Sperm only visible if testes are cut but not by pressure |
| Mature | Active | Spawning capable | • Large ovaries, and located in almost all the abdominal cavity<br>• Yellow to orange in color<br>• Oocytes can be observed | • Large testes, located in almost all the abdominal cavity<br>• White testes<br>• Sperm visible if testes are pressed |
| Mature | Active | Spawning | • Ovaries with translucent oocytes (hydrated oocytes) | • Similar characteristics as mature phase are observed<br>• Testes increase in volume |
| Mature | Inactive | Post spawning | • Red ovaries<br>• Ovaries decrease in volume | • Red testes<br>• Testes decrease in volume<br>• Sperm not visible if testes are pressed |
| Mature | Inactive | Rest | • Pink to white ovaries<br>• Ovaries completely located along the abdominal cavity<br>• Ovaries decrease in volume | • White to pink testes<br>• Testes fully present along the abdominal cavity<br>• Testes decrease in volume |

Nutritional index ($I_N$) was calculated from a modification of the condition factor or Fulton index (K) proposed by *Nikolsky (1963)*, using the weight without viscera rather than the total weight to correct for the effect of the reproductive state of the organisms (*Granado-Lorencio, 1996*):

$$I_N = \frac{\text{Eviscerated weight}}{\text{Fork length}^3} \cdot 100$$

Length at first maturity ($L_{50}$) by sexes was determined from the ogive of maturity by classes with a $L_F$ of five cm through the fitting to the logistic model proposed by *Bakhayokho (1983)*:

$$P = \frac{1}{1 + e^{-(a+bL_F)}}$$

The fitting process was done by applying a General Linear Model (GLM) with binomial errors (logistical regression) and solved with nonparametric bootstrap through the library INBio 1.2 (*Sampedro, Trujillo & Saínza, 2005*) using R software (*R Core Team, 2013*).
Table 2 **Main histologic components and their cellular stages in gonads of *E. alletteratus* caught from December 2011 to November 2012 in the southwest Gulf of Mexico, adapted from *Saber et al. (2015b)*.**

| Histologic component | Cellular stages | |
|---|---|---|
| | Female | Male |
| Ripe gametes (RG) | • Advanced vitellogenic oocytes<br>• Migratory nucleus oocytes<br>• Hydrated oocytes | • Spermatids<br>• Spermatozoa |
| Unripe gametes (UG) | • Primary growth oocytes<br>• Lipid-stage oocytes<br>• Early vitellogenic oocytes | • Spermatogonia<br>• Spermatocytes |
| Free space (FS) | • Free space inside the gonad | • Free space inside the gonad |
| Vesicular connective tissue (VCT) | • Connective tissue inside the gonad | • Connective tissue inside the gonad |

## Microscopic analysis

For establishing the reproductive cycle of *E. alletteratus*, a total of 155 gonads (58 males and 97 females) were analyzed from samplings received from December 2011 to November 2012. The gonads of each tuna were fixed in Davidson's fixative solution (*Shaw & Batle, 1957*) and sections of 5 μm were histologically processed by cutting and staining with hematoxylin and eosin, adhering to specific staining times for *E. alletteratus* (*Cruz-Castán, Curiel-Ramírez & Meiners-Mandujano, 2014*). The average volume fraction (*Vv*) occupied by different cells was determined by quantitative stereology, using the Weibel microscope reticle (*Weibel, Kistler & Scherle, 1966*) with 42 points and following the methodology described by *Briarty (1975)* that was applied in other aquatic organisms by several authors (*Lowe, Moore & Bayne, 1982*; *Seed & Suchanek, 1992*; *Cáceres-Martínez & Figueras, 1998*; *Curiel-Ramírez & Cáceres-Martínez, 2004*, *2012*). Under the microscope, we conducted five random counts in each histological slide. The main components (*xi*) were identified and classified according to the cellular characteristics detailed in Table 2; the fraction volume occupied by each component were then expressed in percentages with:

$$V_V = \frac{\sum xi}{210} \cdot 100$$

## Sea surface temperature

Monthly data of SST were obtained from the telematics interface for the visualization and analysis of data of "Giovanni" (*Acker & Leptoukh, 2007*) remote perception from a satellite with a spatial resolution of four km from December 2009 to November 2012. A temporal series of the monthly average of the SST was built from a regular polygon that included the area of the captures. The temporal evolution of the SST was contrasted against the $I_G$ of *E. alletteratus* to value the degree of temporal coincidence. To determine if there is

**Table 3 Sample size, length and weight descriptors by sex for *E. alletteratus* caught in the southwest Gulf of Mexico from December 2009 to November 2012.**

| Sex | Number of individuals | Mean $L_F$ (cm) | Mean weight (g) | Range length (cm) | Range weight (g) |
|---|---|---|---|---|---|
| Males | 455 | 41.48 ± 7.98 | 1209.09 ± 996.57 | 28.5–80.7 | 350–8560 |
| Females | 480 | 41.96 ± 7.05 | 1195.02 ± 730.56 | 28.2–68.1 | 340–4520 |
| Undetermined | 16 | 40.02 ± 7.87 | 1003.07 ± 435.68 | 28.9–52.0 | 350–1840 |

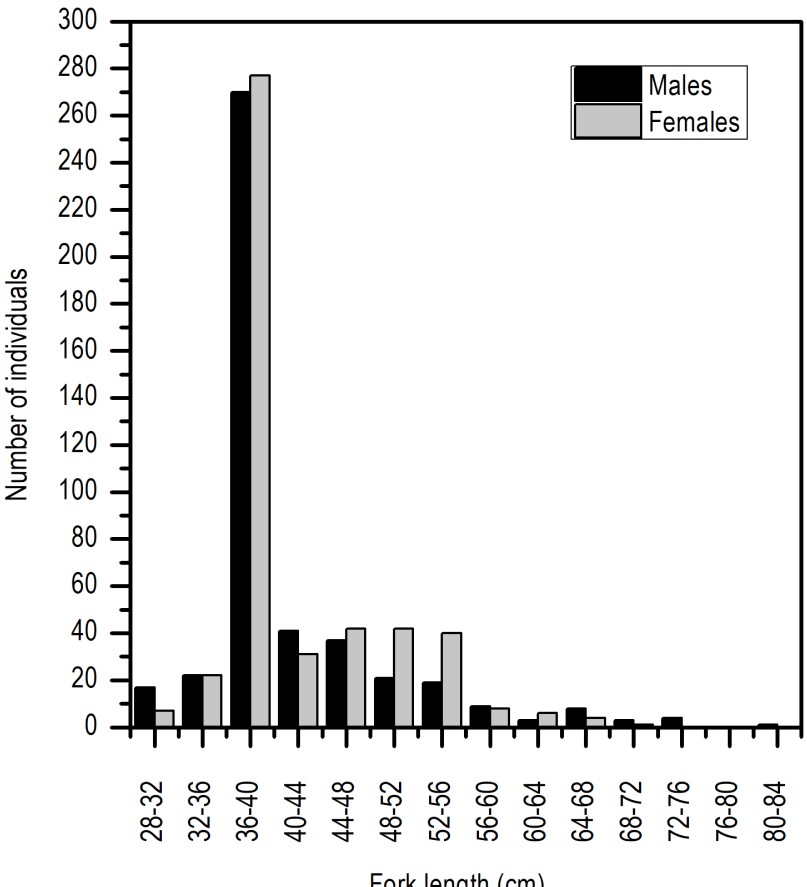

**Figure 2 Length frequency distribution of *E. alletteratus* caught from December 2009 to November 2012 in the southwest Gulf of Mexico.**

a correlation between reproductive activity and temperature, a Pearson correlation analysis was performed.

## RESULTS

A total of 951 organisms were collected, with an average body length of 41.68 cm $L_F$ (+ 7.52 cm) and an average weight of 1196.48 g (+ 861.37 g). Table 3 shows details regarding the body length and weight by sexes.

The distribution of length frequency was strongly skewed towards a length interval of 36–40 cm $L_F$ (Fig. 2).
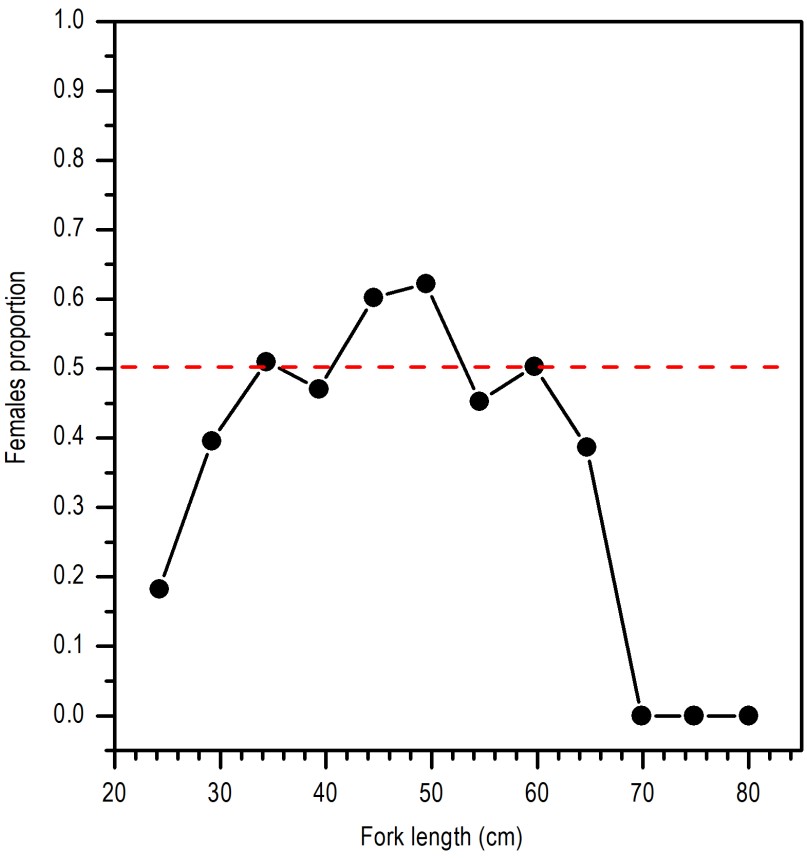

**Figure 3 Sex ratio of *E. alletteratus* caught from December 2009 to 2012 in the southwest Gulf of Mexico.**

Even though the sex ratio of this species is 1:1 in global terms ($\chi^2 = 0.668$, $P > 0.05$), a greater ratio of males in the length range of 28 and 30 cm $L_F$ and above 65 cm of $L_F$ was observed. Females were clearly dominant in the length range of 45 and 50 cm $L_F$ (Fig. 3).

Both for males and females, there was a marked increase in the $I_G$ from April which extended up to September, with fluctuations along these months; maximums were found in April and July and minimums from October to February. In contrast, the $I_H$ was highest in December, decreased notoriously up to March, and increased again from April to September with slight fluctuations. Nutritional index ($I_N$) had highest values in April and October with fluctuations along the rest of the months (Fig. 4).

The length at first sexual maturity was 34.35 cm $L_F$ for males and 34.60 cm $L_F$ for females; in a combined analysis, the length at first maturity for males and females was 34.40 cm $L_F$ (Fig. 5).

Histologically, a temporal progression of the percentage of ripe gametes (RG) at population level (females and males) was observed, with maximum percentages (>60%) in May and July. The RG began to occupy a greater percentage of gonad volume in April, with 49.76% + 5.94%, and decreased abruptly in October. Also, it is important to mention that during May and July, there was a marked decrease of free spaces (FS) in gonads (Fig. 6).

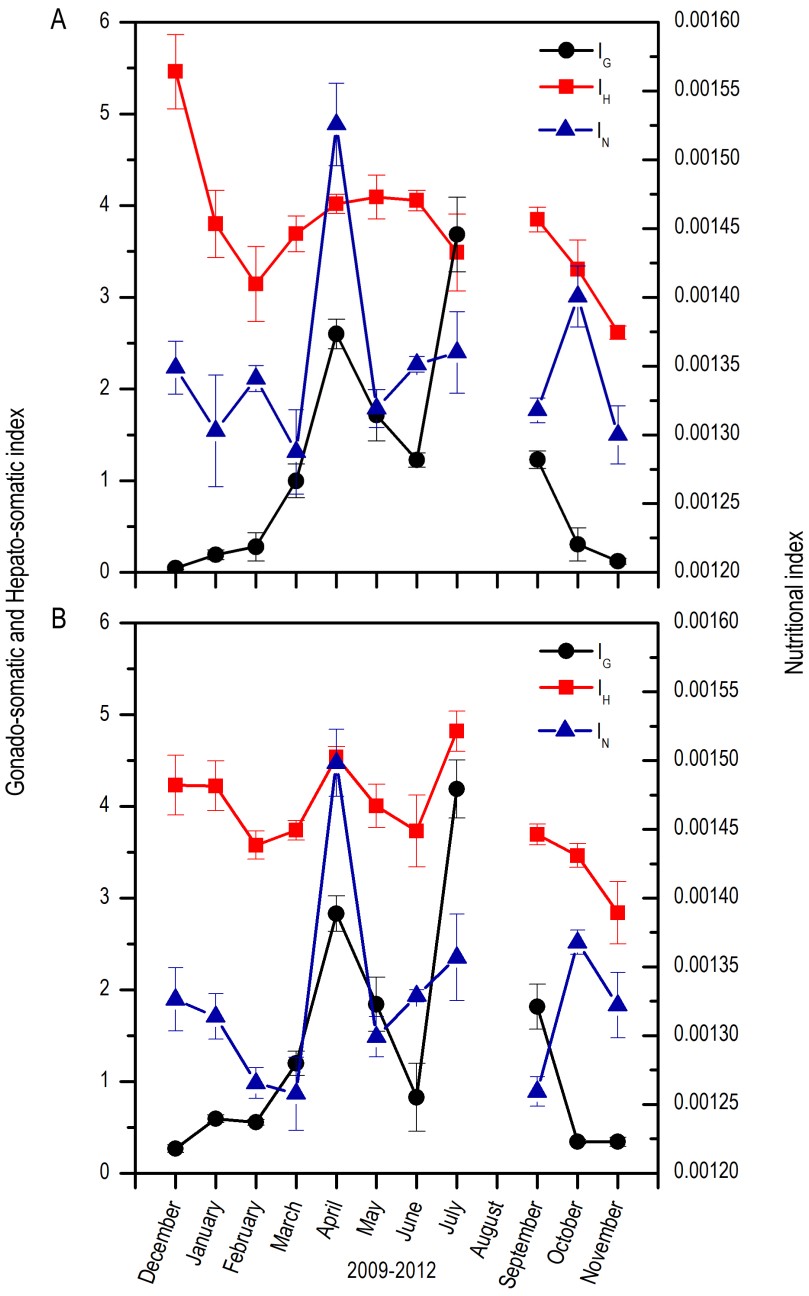

**Figure 4 Monthly changes in the gonadosomatic index ($I_G$), Hepatosomatic index ($I_H$), and nutritional index ($I_N$) for *E. alletteratus* caught from December 2009 to November 2012 in the southwest Gulf of Mexico.** (A) Males. (B) Females.

Comparing the evolution of SST at the study area and the $I_G$, a direct synchrony between the beginning of the increase of gonadic mass and the temperature was observed. At approximately 25 °C, a first maximum of $I_G$ was observed and around 28 °C, the main maximum of $I_G$ was reached; above this temperature, the process was deactivated (Fig. 7). Thus, the temporal evolution in the reproductive activity can be explained with a normal distribution with the majority fraction volume occupied by RG in gonads from April to September ($R^2 = 0.86$) (Fig. 8).

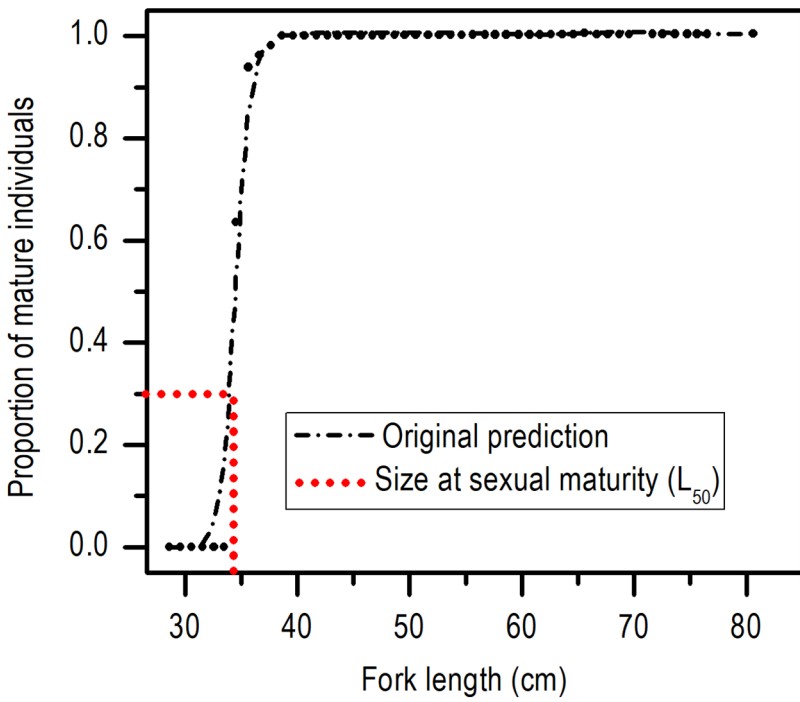

**Figure 5 Size at sexual maturity ($L_{50}$) for females and males of *E. alletteratus* caught from December 2009 to November 2012 in the southwest Gulf of Mexico.**

## DISCUSSION

The maximum $L_F$ of *E. alletteratus* recorded in the current study (80.7 cm) was inferior to those reported by other authors for eastern Atlantic and western Mediterranean, e.g. 85 cm $L_F$ (*Macías et al., 2009*), 97.8 cm $L_F$ (*Hajjej et al., 2010b*), and 84 cm $L_F$ (*Valeiras et al., 2008*). Regarding the minimum length, an individual of 28.2 cm $L_F$ was registered, sensibly smaller than the minimum length reported for specimens sampled in the western Mediterranean, 32–56 cm $L_F$ (*Valeiras et al., 2008*; *Macías et al., 2006*); in the Gulf of Gabes, 34 cm $L_F$ (*Hajjej et al., 2010b*), and in the eastern Atlantic, 31 cm $L_F$ (*Neves dos Santos & García, 2006*). This length difference could be attributed to the selectivity of fishing gears and to the latitudinal gradients in which this species is distributed. In spite of these regional differences, the length interval sampled in this study, is one of the broadest used for studies of reproduction of this species, and it includes specimens very near to the growth asymptote of *E. alletteratus* of the southwest Gulf of Mexico (*Alcaráz-García, 2012*).

Length distribution was multi-modal; however, there was a more notorious mode from 36 to 40 cm $L_F$. The majority of the captures were carried out with gillnets of three inches mesh size, which had an influence on the proportion of capture for each length group. *Macías et al. (2009)* reported an influence of the fishing gear on length distribution of captures along the Spanish coasts. They compared tuna trap nets with coastal sport fishing with trolling and determined multi-modal distributions for both methods which affected different length segments. For the trap nets, the interval of length was wider (62–85 cm $L_F$), without a dominant mode on the rest; in sport fishing, the interval was

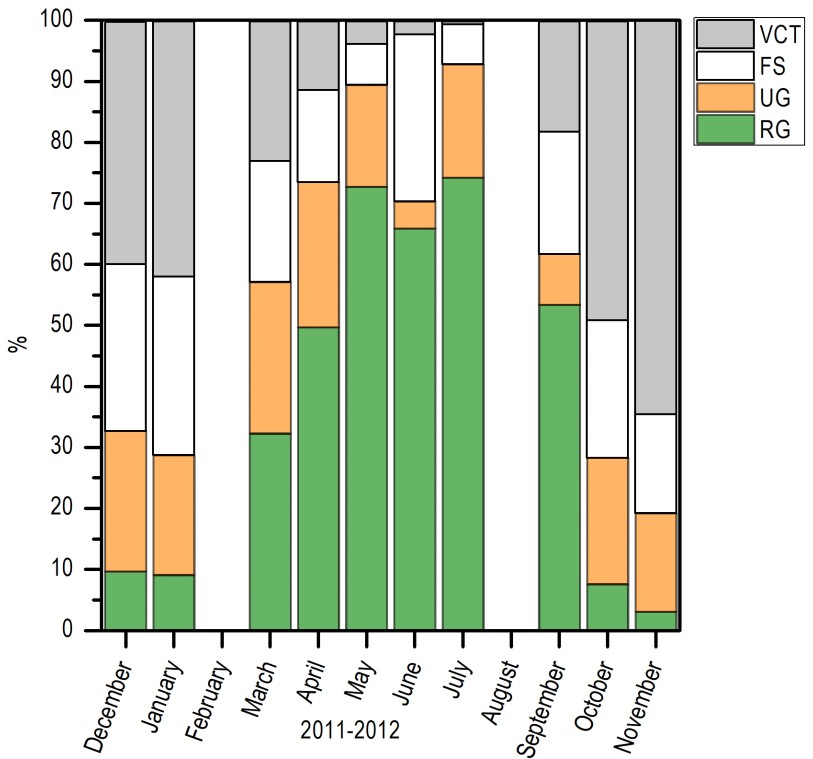

**Figure 6 Variation of the volume percentage of main histologic components of *E. alletteratus* caught from December 2011 to November 2012 in the southwest Gulf of Mexico.** RG, Ripe gametes; UG, Unripe gametes; FS, Free spaces; and VCT, Vesicular connective tissue.

narrower (37–48 cm $L_F$), with a predominant mode (41 cm $L_F$). In this study, the length spectrum was relatively wide, from juveniles to adults, which is a fundamental condition for carrying out a precise description of the reproductive process of any species.

The sex ratio is a sensible indicator of fish populations viability (*Nikolsky, 1963*; *Santamaría-Miranda & Rojas-Herrera, 1997*); a significant difference between the number of females vs males can be attributed to different survival rates or segregated sex distribution (*Lucano-Ramírez, Ruiz-Ramírez & Rojo-Vázquez, 2005*), which results in mortality and growth rates differentiated by sex. In the case of *E. alletteratus* in the southwest Gulf of Mexico, the population in global terms is in sex equilibrium (1:1), and the predominance of larger sized males is due to the fact that males reach greater lengths (*Alcaráz-García, 2012*). This finding is in agreement with other studies in tunids; for example, in albacore (*Thunnus alalunga*) fisheries in the western Mediterranean Sea, the number of females decreased for the greater sizes (*Saber et al., 2015a*).

According to the temporal evolution of $I_G$, the reproductive period of *E. alletteratus* in the southwest Gulf of Mexico starts in April and extends to September. This reproductive period of *E. alletteratus* is similar to the described by *Posada-Peláez et al. (2012)*, who determined that for the Colombian Caribbean region, the first spawning take place in April–June and the second and the most intense, during August–September. On the other

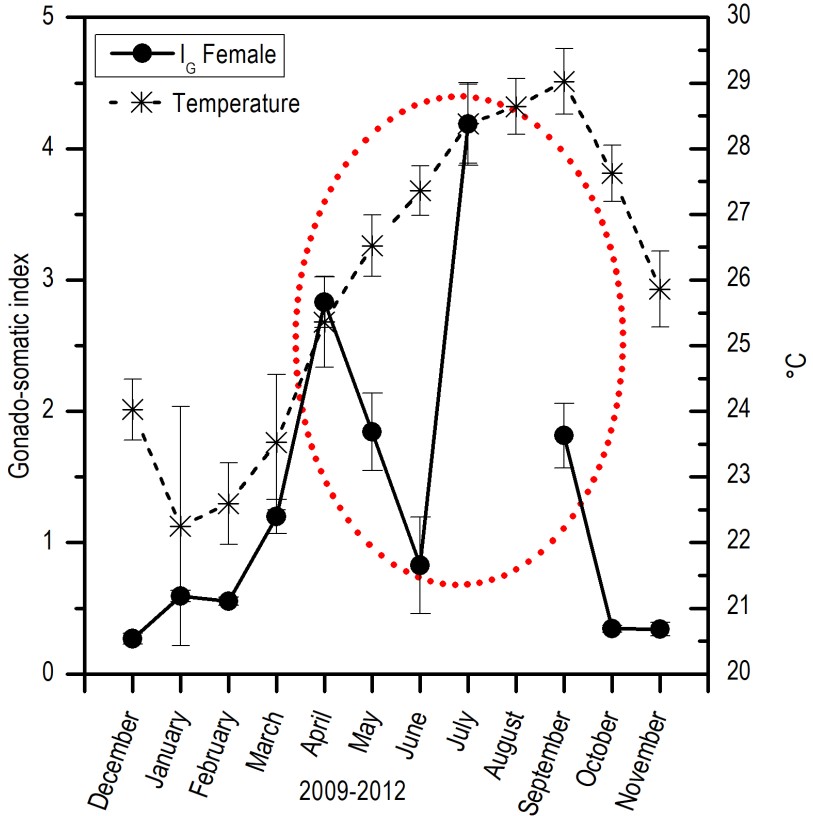

**Figure 7 Relation between monthly changes in the sea surface temperature (SST) and the gonadosomatic index (IG) for *E. alletteratus* caught from December 2009 to November 2012 in the southwest Gulf of Mexico.**

hand, *Diouf (1981)* recorded a reproduction period between July and September along the coasts of Senegal; while in the Mediterranean Sea, both western and eastern, spawning occurs from May to September (*Collette & Nauen, 1983*; *Kahraman et al., 2008*), with an $I_G$ maximum in July, which coincides with the maximum found in this study. However, this study clearly proves the existence of two well-differentiated $I_G$ peaks within an extended reproductive period, which indicates a particular reproduction strategy of *E. alletteratus* in the southwest Gulf of Mexico compared to populations in the eastern Atlantic and Mediterranean Sea where a less extended reproductive peak has been reported. Furthermore, the two $I_G$ peaks were in agreement with the volume fraction occupied by RG in these months. In this sense, histological analysis was used as a complementary study in order to prevent a false conclusion since a decrease in $I_G$ can be attributed to a spawning, but this decrease could be due to collect minor length individuals and a decrease in the gonad weight as a consequence, without been a spawning. The major proportion of FS, connective tissue (VCT) and unripe gametes (UG) correspond with a decrease of reproductive activity for this species.

The relation of variability between $I_H$ and $I_N$ with the development of reproductive activity ($I_G$) showed from December to February, corresponds to the eve of the reproductive period. Besides, an inverse relation between the indexes for both sexes was

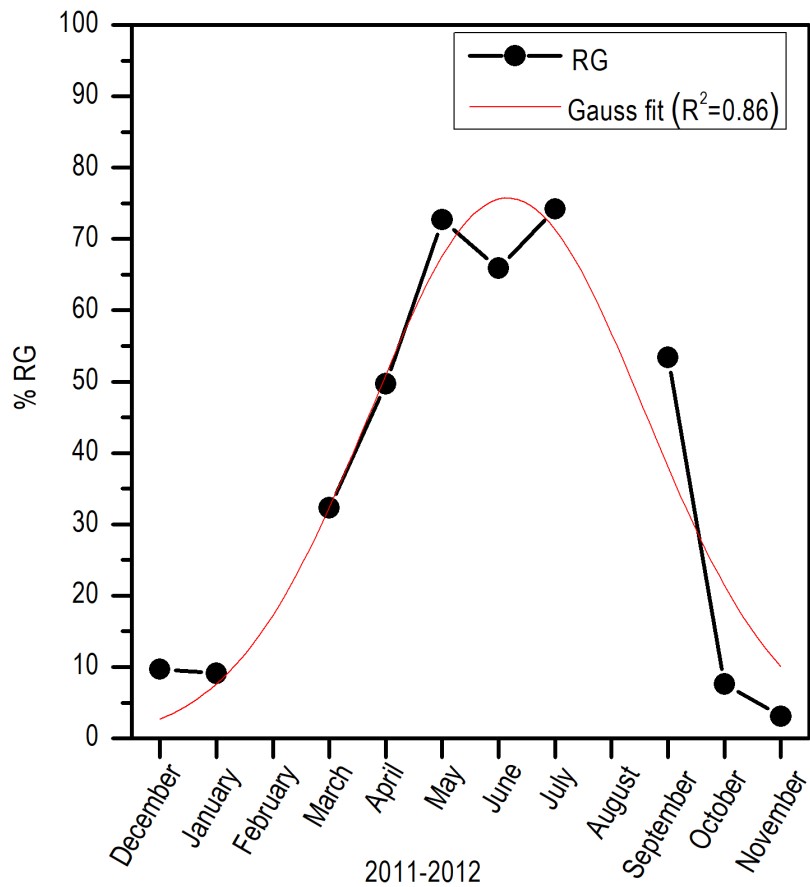

**Figure 8 Monthly variation of ripe gametes percentages of *E. alletteratus* caught from December 2011 to November 2012 in the southwest Gulf of Mexico.** The red line represents the relationship fitted to a Gaussian model.

observed, which suggests the relocation of energy resources from the liver and fatty bodies towards the reproductive system (*Arellano-Martínez et al., 2001*). However, during the reproductive period from April to September, with two clearly defined peaks in April and July, a direct relationship between $I_H$, $I_G$ $I_N$ was observed, suggesting that there was no evident energetic relocation and that the energy for reproductive activity was provided by immediate feeding. During this period, tunas specialize in a diet rich in the engraulid, *Anchoa hepesetus* (*Bouchot-Alegria, 2012*), a species with high fatty content, which enables an increase in reproductive activity without the need of relocating energy from the liver or the muscular tissues. Moreover, after the two maximum $I_G$ peaks, the $I_N$ decreased, which suggests periods without feedings, with immediate recoveries for a following spawning or energy accumulation for the following reproductive season. This could also be demonstrated by the existence of constant availability of food, which allows the organisms to maintain beneficial physiological conditions during the spawning and the post-spawning period (*Acevedo et al., 2007*).

No significant differences between the length at first sexual maturity of males (34.35 cm of $L_F$) and females (34.60 cm $L_F$) were found; therefore, it is adequate to use a length at first sexual maturity of the population (34.40 cm $L_F$). According to the growth estimates

of *E. alletteratus* of the southwest Gulf of Mexico (*Alcaráz-García, 2012*), these sizes correspond to an age of approximately two years. This data is relevant since, although the length at first sexual maturity of this study strongly differed compared to estimates of the western Mediterranean Sea (56 cm $L_F$: *Valeiras & Abad, 2006*), the Gulf of Cadiz (57 cm $L_F$: *Rodríguez-Roda, 1966*), the Gulf of Guinea (~43 cm $L_T$: *Chur, 1973*), and the coasts of Senegal (40 cm $L_T$: *Diouf, 1981*), sex maturity occurs at about two years of age. These results demonstrate that *E. alletteratus* specimens of the southwest Gulf of Mexico reproduce at the same age like the rest of the populations of the eastern Atlantic, but at a length between 15% and 65% smaller, which coincides with the hypothesis that tropical fish tend to be smaller and with wider reproductive periods than those of greater latitudes (*Kokita, 2004*; *Watt, Mitchell & Salewsky, 2010*; *Weber et al., 2015*) or zones of higher biological productivity (*Geist, 1987*; *Garvey & Marschall, 2003*).

For *E. alletteratus* in the southwest Gulf of Mexico, it was observed that the changes in the gonad ($I_G$ and the increase in RG) that unchain the reproductive period start when SST rises to approximately 24–25 °C, reaching their maximum activity at about 28 °C and decline abruptly above this threshold. This means that the optimum thermal window for *E. alletteratus* reproduction in the southwest Gulf of Mexico is asymmetric, skewed towards high temperatures, and occurs in a temperature range from 24 to 28 °C; beyond this range, the process is deactivated.

The extended reproductive period for *E. alletteratus* in the southwest Gulf of Mexico (five to six months), compared to populations in the eastern Atlantic (~3.5 months), is due precisely to the temporal extension of the optimum thermal window in each distribution area of this species. *Gunter (1957)* mentioned that the small seasonal variability could increase the possibility of finding specimens in reproduction in any season of the year as a direct response induced by the temperature over the metabolic rate. There is a relation between the span of the reproduction season, the type of spawning, and the latitude; thus, at high latitudes (brief summer period), the fish species have short, massive, and well-defined spawning periods (*Cushing, 1975*; *Blaxter & Hunter, 1982*). However, at lower latitudes (subtropical and tropical areas), periods of reproduction are prolonged and may be limited to a broader season, but with partial spawning, such as we recorded for *E. alletteratus* in the southwest Gulf of Mexico. In some cases, it may last the whole year (*Cushing, 1975*).

## CONCLUSIONS

*Euthynnus alletteratus* distributed in the southwest Gulf of Mexico has an extensive reproductive period of six months, lasting from April to September, with plausible evidence of two peaks that show the increase in reproductive activity, one occurs in April and the main one in July. *E. alletteratus* reaches its sexual maturity at 34.40 cm $L_F$, without significant differences between sexes. However, although the length at first sexual maturity is smaller than in the eastern Atlantic, the age of first maturity of *E. alletteratus* in the southwest Gulf of Mexico is around 2 years and therefore similar to that in the eastern region. Finally, the optimum thermal window for the reproduction

of this species ranges from 24 to 28 °C, determining the temporal extension of the reproductive period.

Understanding the reproductive biology of the species is a crucial aspect to provide solid scientific knowledge for fisheries management. In this sense, the results of this study allowed to get for the first time a detailed view about the reproductive dynamics of *E. alletteratus* inhabits the Gulf of Mexico. This key information will support future assessments for this species in Mexican waters, and allow us to design new strategies for its proper management.

## ACKNOWLEDGEMENTS

The authors would like to recognize effort and support to all fishermen of Antón Lizardo, Veracruz, México. The authors also thank M.C. Magnolia Salcedo Garduño for her help in processing samples and M.C. Ana Gabriela Galicia Cruz for her support in graphics edition. Analyses and visualizations used in this paper were produced with the Giovanni online data system, developed and maintained by the NASA GES DISC.

### Funding

The main financial support was provided by the research project PROMEP/103.5/09/4482 "Aspectos de la dinámica poblacional y ecológica de *E. alletteratus* (Rafinesque, 1810) capturado en el Sistema Arrecifal Veracuzano"—Mexican Ministry of Public Education (SEP). Roberto Cruz-Castán received financial support from CONACYT postgraduate scholarship No. 318721. The study was also supported by grant of project No. 5249.14 DGEST to Sergio Curiel. There was no additional external funding received for this study. The funders had no role in study design, data collection and analysis, decision to publish, or preparation of the manuscript.

### Grant Disclosures

The following grant information was disclosed by the authors:
PROMEP/103.5/09/4482 "Aspectos de la dinámica poblacional y ecológica de *E. alletteratus* (Rafinesque, 1810) capturado en el Sistema Arrecifal Veracuzano"—Mexican Ministry of Public Education (SEP).
CONACYT postgraduate scholarship: 318721.
Project No. 5249.14 DGEST.

### Competing Interests

The authors declare that they have no competing interests.

### Author Contributions

- Roberto Cruz-Castán conceived and designed the experiments, performed the experiments, analyzed the data, contributed reagents/materials/analysis tools, prepared figures and/or tables, authored or reviewed drafts of the paper, approved the final draft.

- César Meiners-Mandujano conceived and designed the experiments, performed the experiments, analyzed the data, contributed reagents/materials/analysis tools, authored or reviewed drafts of the paper, approved the final draft.
- David Macías conceived and designed the experiments, performed the experiments, analyzed the data, contributed reagents/materials/analysis tools, authored or reviewed drafts of the paper, approved the final draft, classification of oocyte stages.
- Lourdes Jiménez-Badillo conceived and designed the experiments, performed the experiments, analyzed the data, contributed reagents/materials/analysis tools, authored or reviewed drafts of the paper, approved the final draft.
- Sergio Curiel-Ramírez conceived and designed the experiments, performed the experiments, analyzed the data, contributed reagents/materials/analysis tools, authored or reviewed drafts of the paper, approved the final draft.

### Data Availability

Raw data are available in the Supplemental Materials.

### Supplemental Information

Supplemental information for this article can be found online at http://dx.doi.org/10.7717/peerj.6558#supplemental-information.

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
