# Peer review of "Reproductive biology of little tunny Euthynnus alletteratus (Rafinesque, 1810) in the southwest Gulf of Mexico"

_PeerJ, doi:10.7717/peerj.6558_

## Round 0.1 · original submission · Minor Revisions

· Academic Editor

Minor Revisions

Please respond to the reviewers' comments, including those on the annotated manuscript provided by reviewer 2. Also, please double check the Excel file containing raw data to ensure that it opens properly and contains a list of abbreviations as recommended by reviewer 3.

·

Basic reporting

no comment...

Experimental design

no comment...

Validity of the findings

no comment...

Additional comments

The manuscript provides some important parameters concerning with the fisheries management of E. alletteratus having a commercial value in Mexican fisheries industry. The article also has the powerful data set as well as the references. However, in terms of fisheries management of little tunny in Mexican waters, a final paragraph referring the importance of the study should be added to the end of the Conclusion section. After that, it can be published.

Reviewer 2 ·

Basic reporting

no comment

Experimental design

no comment

Validity of the findings

no comment

Additional comments

I reported my suggestions to authors on the text

Annotated reviews are not available for download in order to protect the identity of reviewers who chose to remain anonymous.

·

Basic reporting

Given that I am not a native of English I will refrain from style or grammar comments. In terms of the effectiveness of communication, the message comes across without doubt. Writing is clear and technically correct. References are numerous, perhaps too many. Background is properly given. The article structure conforms professional and customary ways. Raw data are at disposal but I had some problems opening the Excel file. I got messages that the file had to be repaired. Once open the data base of the article can be seen. I missed in the Excel a glossary of abbreviations. Yes, you can find them in the text but it is easier for a reader to have them in the same file. I found a good correspondence between objectives, methods and results. Conclusions are correctly derived from observations plus analysis.

Experimental design

Methods used are standard. The histology analysis complemented well and was consistent with visual assessment of individual’s maturity. I miss, however, a clearer statement on up to which gonadal phase the individuals are considered mature (your Table 1). I have seen in literature that stages “post spawning” and “rest” are viewed as “not mature” despite of the fact that evidently such individuals area beyond their first sexual maturity.

Validity of the findings

The topic treated is interesting and valuable as criteria for management of the fish species concerned plus implications for the general biology of tropical fishes, for instance, the corroboration of extended but defined reproductive periods and environmental correlates, i.e., sea temperature. On the other hand, it would have been better to have sampled the population independently from the fishery because fishers operate in a directed manner and as it is of common knowledge, estimation of things like the first sexual maturity are strongly depend of the size range at hand. Nevertheless, the knowledge gained through the study is useful and relevant and I understand that in practice what a scientist can do in the field has a number of limitations, for instance, funding

---

## Round 0.2 · accepted · Accept

· Academic Editor

Accept

Thank you for addressing the reviewers' comments and improving the manuscript accordingly.

#